# Development of a Fully Autonomous Offline Assistive System for Visually Impaired Individuals: A Privacy-First Approach

**DOI:** 10.3390/s25196006

**Published:** 2025-09-29

**Authors:** Fitsum Yebeka Mekonnen, Mohammad F. Al Bataineh, Dana Abu Abdoun, Ahmed Serag, Kena Teshale Tamiru, Winner Abula, Simon Darota

**Affiliations:** 1Department of Mechanical and Aerospace Engineering, United Arab Emirates University, Al Ain 15551, United Arab Emirates; 201950306@uaeu.ac.ae (F.Y.M.);; 2Aerospace Department, Khalifa University, Abu Dhabi 127788, United Arab Emirates; 3Electrical and Communication Engineering Department, United Arab Emirates University, Al Ain 15551, United Arab Emirates; 4Department of Computer Science and Software Engineering, United Arab Emirates University, Al Ain 15551, United Arab Emirates

**Keywords:** assistive technology, object detection, OCR, face recognition, voice recognition, offline AI, Raspberry Pi

## Abstract

Visual impairment affects millions worldwide, creating significant barriers to environmental interaction and independence. Existing assistive technologies often rely on cloud-based processing, raising privacy concerns and limiting accessibility in resource-constrained environments. This paper explores the integration and potential of open-source AI models in developing a fully offline assistive system that can be locally set up and operated to support visually impaired individuals. Built on a Raspberry Pi 5, the system combines real-time object detection (YOLOv8), optical character recognition (Tesseract), face recognition with voice-guided registration, and offline voice command control (VOSK), delivering hands-free multimodal interaction without dependence on cloud infrastructure. Audio feedback is generated using Piper for real-time environmental awareness. Designed to prioritize user privacy, low latency, and affordability, the platform demonstrates that effective assistive functionality can be achieved using only open-source tools on low-power edge hardware. Evaluation results in controlled conditions show 75–90% detection and recognition accuracies, with sub-second response times, confirming the feasibility of deploying such systems in privacy-sensitive or resource-constrained environments.

## 1. Introduction

Visual impairment represents a significant global health challenge, affecting approximately 285 million people worldwide, with 39 million experiencing complete blindness [1]. These individuals face substantial barriers in navigating their environments, accessing textual information, and maintaining social connections [2]. Visually impaired individuals often encounter significant barriers in their efforts to interact with and interpret their environments. Recent advances in artificial intelligence and edge computing have opened new possibilities for developing sophisticated assistive technologies that can operate independently of cloud infrastructure [3,4].

Traditional assistive technologies, including screen readers, guide canes, and magnification devices, provide basic functionality but lack the intelligent interpretation capabilities offered by modern AI systems [5]. Commercial solutions such as Microsoft Seeing AI and Google Lookout leverage cloud-based artificial intelligence to provide advanced object recognition and scene description [6]. Although assistive technologies have evolved to include screen readers, smart navigation tools, and AI-enabled devices, many of these solutions are dependent on cloud-based infrastructures. While cloud platforms offer advanced computational resources and high recognition accuracy, they come with critical drawbacks such as data privacy concerns, latency in communication, constant internet dependency, and higher operational costs.

The privacy implications of cloud-based assistive systems are particularly concerning, as they often require transmission of sensitive personal data, including images of the user’s environment and biometric information [7]. Furthermore, the reliance on constant internet connectivity limits their applicability in rural areas, developing regions, or situations where network access is unreliable [8].

To overcome these limitations, this research proposes a fully offline assistive system built on a Raspberry Pi 5. The system integrates core AI functionalities—including object detection, optical character recognition (OCR), face recognition, and voice-command processing—executed entirely on-device. This architecture supports real-time interaction without requiring cloud access, thereby enhancing user privacy, reducing latency, and extending accessibility to remote or low-resource settings. The approach bridges the gap between affordability and autonomy in modern assistive technologies, addressing the critical need for privacy-preserving, accessible solutions [9].

The primary aim of this study is to design and implement a Python-based assistive platform that provides real-time visual interpretation and voice interaction capabilities for visually impaired users, all while operating independently of internet connectivity. The specific objectives of the system are as follows:Develop a lightweight object detection module to identify and localize items in the user’s environment.Integrate an OCR engine to extract printed text and convert it to speech, enabling access to textual content.Implement a face recognition system that can identify pre-registered individuals and communicate their identity to the user.Design a voice-command interface that enables hands-free control over the system’s functionalities.Ensure optimized performance on a Raspberry Pi 5, maintaining responsiveness without relying on external computing resources.Demonstrate a privacy-first approach with comprehensive local data processing and zero external data transmission.

This work contributes significantly to the field of assistive technology by introducing an affordable, portable, and privacy-conscious solution tailored for visually impaired users. The key contributions include (1) a novel comprehensive offline multimodal assistive system integrating object detection, OCR, face recognition, and voice control on a single edge device; (2) a systematic evaluation of open-source AI models for embedded assistive applications; (3) novel optimization strategies for achieving real-time performance on resource-constrained hardware; and (4) a privacy-preserving architecture that eliminates dependence on cloud services [10]. Unlike commercial systems that require internet-based AI services, the proposed system functions autonomously, making it particularly useful in rural or underdeveloped regions with limited connectivity.

The scope of this project includes the development and evaluation of a real-time, offline assistive system that provides voice-guided support based on visual inputs. The following core functionalities are implemented:**Object Detection:** Identifies objects in the camera’s field of view and delivers positional feedback to the user via audio.**Optical Character Recognition (OCR):** Reads printed or displayed text from scenes and documents, converting it into spoken words.**Face Recognition:** Detects and identifies individuals from a stored database of known faces, announcing their names when recognized.**Voice Command Interface:** Empowers the user to control the system’s operation, toggle features, and switch modes through spoken commands.**Privacy-First Architecture:** Ensures all processing occurs locally with zero data transmission to external servers.

While the system demonstrates effective performance in controlled indoor settings, current limitations include processing constraints of the Raspberry Pi 5, particularly during simultaneous execution of multiple AI models. These constraints may impact real-time performance under high computational loads. Additionally, performance degrades in challenging environmental conditions such as poor lighting or high background noise [11].

The rest of the paper is organized as follows: Section 2 reviews recent advancements and limitations in assistive technologies, providing comprehensive comparison tables and research gap analysis. Section 3 describes the system architecture, hardware components, software stack, and operational workflow, including detailed model selection justification, threading architecture, and privacy-first implementation. Section 4 presents comprehensive testing methodology, evaluation metrics, dynamic scene analysis, and comparative performance analysis with existing systems. Section 5 concludes with key contributions, current limitations, and future research directions.

## 2. Related Work

### 2.1. Overview of Assistive Technologies

The field of assistive technology for visually impaired individuals has evolved significantly over the past decade, driven by advances in computer vision, natural language processing, and edge computing [12]. Early systems relied primarily on simple audio feedback and tactile interfaces, while modern solutions leverage sophisticated AI models for environmental understanding [3].

Recent advancements in artificial intelligence (AI), edge computing, and embedded systems have significantly enhanced assistive technologies, particularly for individuals with visual impairments. These technologies aim to improve users’ autonomy by enabling real-time interaction with their surroundings through audio, visual, and tactile feedback [13].

Cloud-based assistive systems have dominated the market due to their access to powerful computational resources and large-scale datasets. Microsoft Seeing AI [14], Google Lookout [15], and Amazon Alexa’s accessibility features [16] represent the current state-of-the-art in commercial assistive technologies. However, these systems face significant limitations in terms of privacy, connectivity requirements, and cost [17].

As shown in Table 1, current assistive systems.

Historically, tools such as Braille readers, guide canes, and handheld magnifiers offered basic navigation and reading support. With the emergence of AI-powered systems, however, there has been a shift toward intelligent solutions capable of performing real-time object recognition, scene analysis, and speech interaction [20]. Nevertheless, their reliance on cloud connectivity raises privacy concerns, introduces latency, and renders them unusable in bandwidth-constrained or remote settings [21]. The limitations of current approaches, as detailed in Table 2, demonstrate the following:

To address these issues, recent research has focused on the development of offline assistive systems. Dubey et al. [18] demonstrated the feasibility of using Raspberry Pi-based platforms for local data processing, providing a balance between cost, privacy, and usability. However, the limited computational power of such devices poses challenges related to real-time performance and simultaneous execution of multiple functionalities.

#### 2.1.1. Object Detection in Assistive Systems

Object detection is central to improving spatial and environmental awareness for visually impaired users. Deep learning models such as YOLO (You Only Look Once) and MobileNet have been widely adopted in assistive applications for their ability to detect and classify objects quickly and accurately.

The evolution of YOLO architectures has been particularly significant for embedded applications. YOLOv1 through YOLOv8 have shown progressive improvements in both accuracy and computational efficiency [22]. YOLOv8, in particular, introduces architectural innovations such as anchor-free detection and improved feature pyramid networks that make it suitable for edge deployment [23].

Dubey et al. [18] explored the use of YOLO for object detection on the Raspberry Pi, confirming its potential for real-time inference despite limited hardware resources. More recent work by Okolo et al. [19] implemented YOLOv8 on Raspberry Pi for assistive navigation systems, achieving 91.70% average accuracy across nine tested obstacles while demonstrating the feasibility of real-time object detection for visually impaired navigation assistance. Varghese and Sambath [24] provided a comprehensive evaluation of YOLOv8’s enhanced performance and robustness, demonstrating its suitability for real-time object detection applications on resource-constrained devices. A comprehensive review by Hussain [25] traced the evolution of YOLO from YOLOv1 to YOLOv8, highlighting the architectural improvements that make YOLOv8 suitable for embedded applications.

The performance comparison of object detection models on embedded devices, as presented in Table 3, shows the following:

While cloud-hosted object detection models exhibit higher performance due to access to larger datasets and powerful servers, their dependence on constant internet connectivity limits their deployment in offline or privacy-sensitive environments. Thus, optimizing object detection algorithms for edge devices such as the Raspberry Pi continues to be an important area of research.

#### 2.1.2. Optical Character Recognition (OCR) for Text-to-Speech Assistance

OCR technology has undergone significant advancement, with modern engines achieving over 95% accuracy on printed text under optimal conditions [26]. However, embedded OCR systems face unique challenges related to computational constraints and environmental variability [27].

OCR is a key feature in assistive technologies that allows users to access printed or digital text via audio feedback. Commercial OCR platforms like Google Vision API and Microsoft Cognitive Services offer high recognition accuracy, yet they are cloud-dependent and therefore unsuitable for privacy-preserving offline applications [20].

To enable on-device OCR, researchers have utilized open-source engines such as Tesseract OCR, which can be deployed efficiently on low-power hardware. Kumar and Singh [28] developed a Raspberry Pi-based OCR system that captures images and translates recognized text into speech. While promising, the system still struggled with handwritten input and non-English characters. Recent studies have demonstrated significant improvements in Tesseract OCR performance through preprocessing techniques, with researchers achieving substantial accuracy gains on challenging datasets [29].

Lavric et al. [21] further highlighted the importance of multilingual OCR in accommodating diverse user populations. Incorporating AI accelerators like Coral TPU or the Raspberry Pi AI Kit can improve OCR inference speed and accuracy, paving the way for more capable offline assistive reading systems.

#### 2.1.3. Face Recognition for Personalized Assistance

Face recognition in assistive technologies presents unique challenges related to user privacy, computational efficiency, and recognition accuracy under varying conditions [30]. Recent advances in lightweight face recognition models have made edge deployment more feasible [31].

Face recognition technology enhances social interaction for visually impaired users by detecting and announcing known individuals in their environment. While cloud-based services such as Amazon Rekognition and Microsoft Face API provide high recognition accuracy, they involve transferring sensitive biometric data over the internet, raising substantial privacy concerns [13].

To mitigate this, researchers have investigated edge-based face recognition systems. De Freitas et al. [32] proposed an offline face recognition framework deployed on embedded systems, achieving promising accuracy while maintaining user data privacy.

Chang et al. [33] implemented a pedestrian navigation system that incorporates face recognition on edge hardware. Their findings indicated that low-resolution cameras and processing limitations remain significant obstacles to performance. As a result, many recent systems leverage OpenCV-based models fine-tuned for Raspberry Pi to achieve an acceptable balance between performance and resource usage.

#### 2.1.4. Voice-Controlled Systems in Assistive Technology

Offline voice recognition has emerged as a critical component for privacy-preserving assistive systems. Modern offline ASR (Automatic Speech Recognition) systems like VOSK and SpeechRecognition achieve competitive accuracy while maintaining user privacy [34].

Voice control is an essential interaction modality for hands-free operation of assistive devices. Commercial solutions such as Amazon Alexa and Google Assistant deliver highly accurate voice recognition but require constant internet connectivity, making them unsuitable for fully offline systems [20].

To enable offline speech recognition, open-source frameworks like VOSK have been developed. VOSK supports real-time transcription on edge devices and enables voice interaction without compromising privacy. Okolo et al. [13] demonstrated that local speech processing systems can achieve comparable accuracy to cloud-based alternatives while offering superior user control and data security.

### 2.2. Research Gap Analysis

Despite significant progress in individual components, several critical gaps remain in current assistive technology research:**Integrated Multimodal Systems:** Most existing research focuses on individual components (object detection, OCR, or face recognition) rather than integrated systems that combine multiple modalities seamlessly.**Privacy-Preserving Architectures:** Limited research exists on comprehensive privacy-first designs that eliminate all external data transmission while maintaining functionality.**Dynamic Environment Testing:** Current evaluations primarily focus on static scenes, with insufficient analysis of system performance in dynamic, real-world environments.**Resource Optimization:** Lack of systematic approaches to optimize multiple AI models for concurrent execution on resource-constrained hardware.**User-Centric Design:** Insufficient focus on actual user needs and preferences in system design and evaluation.

While modern assistive systems have made significant strides in supporting individuals with visual impairments, several fundamental limitations remain. These challenges restrict widespread adoption, especially in low-resource or privacy-sensitive environments:**Cloud Dependency:** Many commercial and academic solutions rely on cloud-based AI processing for functionalities such as object detection, face recognition, and speech processing. This reliance not only limits access in rural or offline scenarios but also introduces serious privacy and data security concerns [13].**Hardware Constraints:** Embedded platforms like the Raspberry Pi offer cost-effective and portable solutions; however, they are often underpowered when executing complex AI models simultaneously, which restricts real-time system responsiveness and accuracy [18].**Multitasking Performance:** Running several AI-based features—such as OCR, face recognition, and object detection—in parallel frequently results in performance degradation. This includes lag, delayed responses, or even failure to recognize user commands [21].

To address these constraints, this study introduces an integrated offline assistive system built entirely on a Raspberry Pi 5. The proposed system consolidates object detection, OCR, face recognition, and voice command handling into a unified and modular Python-based platform. By leveraging edge computing principles and open-source libraries, the system prioritizes privacy, responsiveness, and affordability—making it a practical solution for both urban and underserved regions. This approach aligns with recent research emphasizing the importance of privacy-preserving edge computing architectures for sensitive applications [35].

## 3. System Design and Methodology

### 3.1. System Overview

The conceptual framework of our offline assistive system is illustrated in Figure 1.

The proposed system is a fully offline, Python-based assistive platform designed to empower visually impaired individuals by facilitating interactive engagement with their surroundings. It integrates real-time object detection, optical character recognition (OCR), face recognition, and voice-based control—all implemented locally on a Raspberry Pi 5 without any dependence on cloud infrastructure.

The architecture adopts a modular structure, enabling each core function to operate independently or in combination depending on the selected mode. Voice commands serve as the primary user interface, while auditory feedback ensures seamless and intuitive interaction. The overall workflow is optimized for minimal latency and high usability, allowing the user to switch modes dynamically without manual intervention—ideal for hands-free operation in real-world settings.

### 3.2. Hardware Components

The physical implementation of the system is based on low-cost, energy-efficient hardware components that are readily available and easy to integrate. Table 4 summarizes the main hardware modules used in building the prototype.

### 3.3. Software Stack and Main Libraries Used

The entire system is developed using Python 3.11.2, which offers high flexibility and support for numerous open-source libraries. The software stack incorporates a range of tools tailored for computer vision, speech processing, multithreading, and hardware interfacing. Table 5 provides a detailed overview of the main libraries and tools used across various system functionalities.

To achieve real-time performance on a resource-constrained platform like the Raspberry Pi 5, the system integrates lightweight, open-source AI models selected through an iterative design and testing process. The model selection process involved comprehensive benchmarking of various architectures under different computational constraints, as detailed in Section 3.4.

While Tesseract OCR was initially considered for broader visual interpretation, its capabilities are limited to text recognition and do not extend to general object detection. Moreover, its integration with hardware accelerators such as the Hailo AI module in the Raspberry Pi AI Kit is not natively supported and requires intermediate translation layers or model conversion steps, which introduce additional complexity. In contrast, YOLOv8 provides a robust and flexible object detection framework with direct compatibility for hardware acceleration pipelines.

### 3.4. Model Selection and Justification

The selection of AI models for the offline assistive system required careful consideration of the trade-offs between accuracy, computational efficiency, and real-time performance constraints. This section provides detailed justification for each model choice based on systematic evaluation and benchmarking.

#### 3.4.1. Object Detection Model Selection

The comparison of object detection models on Raspberry Pi 5, as shown in Table 6, demonstrates the following:

From an embedded vision theory perspective, the choice between YOLOv8’s cross-window attention mechanism and MobileNet’s depthwise separable convolution requires careful analysis. While MobileNet architectures traditionally excel in mobile deployment due to their lightweight design, YOLOv8’s architectural innovations provide several advantages for the Raspberry Pi 5 platform:**Memory Access Patterns:** YOLOv8’s unified architecture reduces memory fragmentation compared to MobileNet’s sequential depthwise and pointwise convolutions.**Cache Efficiency:** The Raspberry Pi 5’s ARM Cortex-A76 architecture benefits from YOLOv8’s optimized tensor operations and reduced memory bandwidth requirements.**Computational Complexity:** While MobileNet reduces FLOPs through separable convolutions, YOLOv8’s anchor-free design eliminates post-processing overhead, resulting in better overall performance on ARM architectures.

Empirical testing confirmed that YOLOv8n achieves superior end-to-end performance (800 ms vs. 950 ms for MobileNet-SSD) despite slightly higher theoretical computational requirements.

#### 3.4.2. Face Recognition Architecture

The face recognition encoding optimization analysis, presented in Table 7, shows the following:

The optimal threshold for face encoding storage was determined through systematic analysis balancing recognition accuracy, storage cost, and matching efficiency. Mathematical analysis shows that recognition accuracy follows a logarithmic improvement curve:(1)$$Accuracy\left(n\right)=A_{max}\cdot \left(1-e^{-\lambda n}\right)$$
where *n* is the number of encodings, $A_{max}=85\%$ is the theoretical maximum accuracy, and λ=0.32 is the learning rate parameter. The optimal threshold of five encodings represents the point where marginal accuracy gains (<1%) no longer justify the linear increase in storage and computational overhead.

### 3.5. System Threading Architecture

The system threading architecture and priority management is illustrated in Figure 2.

The system employs a sophisticated multi-threaded architecture with priority-based task scheduling to ensure responsive interaction while managing computational constraints. The threading system is designed based on the real-time interaction needs of visually impaired users, where immediate response to voice commands is critical for system usability.


**Priority Allocation Rationale:**
**Priority 1—Voice Commands:** Highest priority ensures immediate system responsiveness to user instructions, critical for hands-free operation.**Priority 2—Face Recognition:** High priority for social interaction support, enabling timely identification of approaching individuals.**Priority 3—Object Detection:** Moderate priority for environmental awareness, providing continuous but non-critical spatial information.**Priority 4—OCR Processing:** Lower priority for text reading tasks, which can tolerate slight delays without affecting user experience.


**Queue Management:** The system implements a priority queue with 50-command capacity. When the queue reaches capacity, the oldest low-priority commands are discarded to prevent system overload while preserving critical user interactions.

### 3.6. Data Processing and Optimization

Given the limited computational resources of the Raspberry Pi 5, a series of optimization techniques were carefully implemented to ensure that the assistive system delivers responsive, accurate, and real-time performance. These strategies encompass intelligent workload distribution, algorithmic simplification, and efficient resource utilization across various subsystems, including speech recognition, computer vision, and audio synthesis.

#### 3.6.1. Managing Processing Load

To prevent system slowdowns and maintain smooth operation under multitasking conditions, several mechanisms were employed to manage computational load:**Multithreading:** The system leverages concurrent threads to enable the parallel execution of critical functions such as voice command processing, image acquisition, and AI inference. This allows for responsive interaction and seamless switching between modes without significant delays.**Queue-Based Command Handling:** A first-in-first-out (FIFO) queueing mechanism ensures that user commands are processed in the order they are received. This structured handling avoids command overlaps and potential system bottlenecks, particularly under high-demand scenarios.**Optimized Model Execution:** AI models used for object detection (YOLOv8) and face recognition are configured to run at lower input resolutions. This significantly reduces the computational burden while preserving acceptable levels of detection accuracy and robustness in practical use cases.

#### 3.6.2. Improving Speech Recognition Accuracy

Voice interaction is a central feature of the system, requiring precise recognition even in less-than-ideal acoustic environments. To this end, audio input is pre-processed using the following enhancements:**Noise Reduction:** The integration of the SpeexDSP library allows for real-time suppression of background noise, which is critical for achieving clarity in user speech input, especially in dynamic or noisy settings.**Audio Pre-Processing:** At system initialization, a sample of ambient noise is recorded to serve as a reference. This allows the system to better differentiate between user commands and background sounds, improving speech-to-text conversion accuracy during runtime.

### 3.7. Privacy-First Architecture and Implementation

The system implements a comprehensive privacy-first approach that ensures complete data sovereignty and eliminates external dependencies. This architecture addresses growing concerns about biometric data privacy and personal information security in assistive technologies.

#### 3.7.1. Privacy Implementation Details

The privacy-first implementation metrics are detailed in Table 8.

#### 3.7.2. Data Flow Security Analysis

The privacy-first data flow architecture is shown in Figure 3.

The privacy-first architecture ensures that all sensitive data remains within the user’s control:**Zero External Communication:** The system is designed with no network interfaces active during operation, preventing any accidental data transmission.**Ephemeral Processing:** Camera frames and audio samples are processed in memory and immediately discarded, leaving no persistent traces.**Encrypted Local Storage:** Face recognition encodings are stored using AES-256 encryption with user-controlled keys.**Audit Trail:** Complete system operation logging enables users to verify privacy compliance.

### 3.8. System Workflow and Operational Methodology

The offline assistive system is designed to facilitate seamless interaction between visually impaired users and their surroundings by leveraging voice commands and AI-based perception. The system executes a well-structured sequence of operations to deliver real-time feedback and support. A graphical representation of the overall workflow is provided in Figure 4, and the detailed steps are explained below.

#### Explanation of System Workflow

The assistive system operates in a structured sequence of steps to enable visually impaired users to interact with their environment through voice commands.

**Step 1: System Initialization** Upon startup, the system initializes essential hardware components and software models:(i)Activating the Raspberry Pi 5’s camera module and microphone for continuous multimedia input.(ii)Loading pre-trained AI models, including YOLOv8 for object detection and Tesseract OCR for text extraction.(iii)Running SpeexDSP to capture baseline ambient noise for dynamic noise filtering.

**Step 2: Voice Command Monitoring and Mode Activation** The system continuously monitors audio input via the microphone, using VOSK speech-to-text engine to process incoming speech and interpret predefined voice commands such as “Activate”, “Register”, or “Exit".

**Step 3: Combined Detection Mode** In combined mode, the system executes multiple detection tasks in parallel including object detection using YOLOv8, optical character recognition for text extraction, and face recognition against stored encodings.

**Step 4: Face Registration Workflow** The face registration feature enables users to enroll new individuals by prompting for names via voice interaction and capturing face images for encoding storage.

**Step 5: Real-Time Audio Feedback** All results are communicated through Piper Text-to-Speech synthesis, with Pyttsx3 as a lightweight fallback option when system resources are constrained.

## 4. Testing and Evaluation

This section presents a comprehensive evaluation of the proposed offline assistive system in terms of its accuracy, responsiveness, and overall usability. The system was tested under varying conditions, and its performance was benchmarked against key metrics. Additionally, a comparative analysis was conducted with cloud-based AI solutions to highlight the strengths and limitations of an offline deployment.

### 4.1. Testing Conditions and Methodology

To ensure a realistic and rigorous evaluation, the system was subjected to various operational scenarios replicating practical usage by visually impaired individuals:The system was tested indoors across different lighting environments, including well-lit and dim settings, to assess the robustness of vision-based tasks.Voice command performance was measured in both quiet and noisy conditions to simulate real-world acoustic variability.System performance was analyzed under varying computational loads—ranging from the execution of a single AI process to concurrent execution of multiple tasks (e.g., object detection, face recognition, and OCR simultaneously).

### 4.2. Dynamic Scene Evaluation

To address the limitation of static scene testing, comprehensive dynamic scene evaluation was conducted to assess system performance under realistic movement conditions.

#### Dynamic Testing Methodology

Dynamic testing involved recording real-time videos of users walking at different speeds while the system performed object detection, OCR, and face recognition tasks. The testing protocol included the following:(i)**Speed Variations:** Testing at 0.5 m/s (slow walking) and 1.0 m/s (normal walking) to simulate typical user movement patterns.(ii)**Motion Blur Analysis:** Evaluating the impact of camera shake and object motion on detection accuracy.(iii)**Tracking Performance:** Assessing the system’s ability to maintain object identification across consecutive frames.

The static vs dynamic performance comparison is presented in Table 9.

The dynamic testing revealed that while performance degrades with movement speed, the system maintains acceptable functionality for typical user scenarios. Motion blur primarily affects OCR accuracy, while object detection shows greater robustness to movement.

### 4.3. Performance Metrics and Statistical Analysis

Performance assessment focused on key metrics including detection accuracy, recognition rates, and response time across each major functionality. Statistical significance testing was conducted using paired *t*-tests with *p* < 0.05 threshold. Table 10 and Table 11 summarize the system’s quantitative evaluation results.

### 4.4. Training and Validation Analysis

The training and validation curves for YOLOv8 fine-tuning on our assistive dataset are shown in Figure 5.

The training process involved fine-tuning YOLOv8n on a custom dataset of 2500 images relevant to assistive scenarios, including indoor objects, text documents, and human faces. The convergence analysis shows stable training with minimal overfitting, validating the model’s suitability for the target application.

### 4.5. Confusion Matrix Analysis

The confusion matrix for object detection performance is presented in Figure 6.

The confusion matrix analysis reveals strong diagonal dominance, indicating good class separation with minimal cross-class confusion. The primary confusion occurs between structurally similar objects (chair/table), which is expected given the resolution constraints of the embedded system.

### 4.6. Comparison with Previous Work

Statistical significance testing (paired *t*-test, *p* < 0.05) confirms that our system achieves significantly better performance compared to previous embedded assistive systems, particularly in terms of integrated functionality and response time.

The performance comparison with previous work is presented in Table 12.

### 4.7. High-Load Performance Analysis

To evaluate system performance under demanding conditions, high-load scenarios were simulated where object detection, OCR, and face recognition were triggered simultaneously. The analysis compared response delays with and without priority queue scheduling, as shown in Table 13.

The priority queue scheduling demonstrates significant improvements, particularly for time-critical voice command processing, validating the threading architecture design.

### 4.8. Comparison with Cloud-Based Systems

To further contextualize system performance, a qualitative comparison was conducted between the proposed offline solution and standard cloud-based AI systems. This analysis considered aspects such as computational efficiency, latency, user privacy, and deployment flexibility.

The performance comparison between cloud-based and offline systems is shown in Table 14.

The comparative analysis reveals distinct advantages for each approach. The proposed offline system demonstrates complete privacy preservation (10/10) by processing all data locally without external transmission, ensuring full offline capability (10/10) that maintains functionality regardless of internet connectivity. Additionally, the system offers low cost implementation (9/10) through efficient use of readily available hardware components and easy deployment (9/10) with minimal technical expertise required for setup and maintenance.

The comparative evaluation reveals several critical considerations that inform the selection between cloud-based and offline assistive technologies:(i)**Cloud-Based Systems:** Benefit from substantial computational resources, enabling the use of larger and more complex AI models, which enhances accuracy in tasks such as face and object recognition. However, they are inherently dependent on stable internet connectivity, introducing latency and posing privacy concerns when transmitting user data to remote servers.(ii)**Offline Raspberry Pi System:** Prioritizes low-latency, real-time interaction and enhanced user privacy by processing all data locally. While it is limited by hardware constraints, it remains operational without internet access—an essential feature for deployment in rural, low-resource, or privacy-sensitive environments.(iii)**Voice Command Limitations:** Offline voice recognition is comparatively less accurate than cloud-based solutions, particularly in acoustically challenging environments. This is due to the limited size and scope of the onboard language models available for offline use.(iv)**Deployment Flexibility:** The offline solution excels in scenarios where infrastructure is lacking or internet reliability is low, offering a viable, cost-effective alternative to cloud-based assistive technologies.

### 4.9. Implementation Challenges and Solutions

The implementation of a fully offline assistive system on the Raspberry Pi 5 introduced several hardware and software-related challenges. These challenges stem from the need to balance computational demands of deep learning models with real-time performance requirements, all while maintaining usability and robustness in practical environments.

#### Hardware Limitations and Solutions

**Processing Constraints:** The limited computational capacity of the Raspberry Pi 5 makes it difficult to simultaneously execute resource-intensive AI models.


**Solutions Implemented:**
Reduced input resolution for computationally intensive models;Employed multithreading to manage independent tasks concurrently;Introduced queue-based command handling with priority management;Optimized model architectures for ARM processors.


**Camera and Audio Limitations:** Standard Raspberry Pi peripherals showed reduced performance under challenging conditions.


**Solutions Implemented:**
Applied image preprocessing techniques including brightness enhancement;Integrated SpeexDSP noise suppression library;Captured baseline noise profiles for adaptive filtering;Implemented automatic gain control for audio input.


The implementation challenges and solutions summary is presented in Table 15.

## 5. Conclusions

### 5.1. Key Contributions and Findings

This study presents the development of an offline Python-based assistive system designed to enhance the autonomy and accessibility of visually impaired individuals. By integrating object detection, optical character recognition, face recognition, and voice-command capabilities into a compact and affordable Raspberry Pi 5 platform, the system offers a comprehensive, privacy-focused alternative to cloud-dependent assistive technologies.

The key contributions of this research include the following:**Integrated Multimodal System:** First comprehensive offline system combining object detection, OCR, face recognition, and voice control on a single edge device with sub-second response times.**Privacy-First Architecture:** Complete elimination of cloud dependencies with 100% local data processing, addressing critical privacy concerns in assistive technology.**Systematic Optimization:** Novel approach to concurrent AI model execution on resource-constrained hardware through priority-based threading and queue management.**Real-World Validation:** Comprehensive evaluation including dynamic scene testing and statistical significance analysis, demonstrating practical viability.**Open-Source Implementation:** Fully reproducible system using exclusively open-source tools, promoting accessibility and further research.

Through the use of open-source libraries and careful optimization strategies—including multithreading, queue-based task management, and resolution adjustments—the system achieves functional real-time performance within the constraints of limited hardware resources. Evaluation results demonstrate promising accuracy and usability across all core functionalities, particularly in controlled indoor environments. Notably, the system maintains high levels of data privacy and responsiveness without the need for internet connectivity, making it especially suitable for deployment in low-resource or remote settings.

Dynamic scene testing revealed that while performance degrades with user movement (15–18% accuracy reduction at normal walking speed), the system maintains acceptable functionality for typical use scenarios. The priority-based threading architecture demonstrated significant improvements in system responsiveness, with 71% faster voice command processing under high-load conditions.

### 5.2. Limitations and Future Work

Despite its strengths, the current implementation faces several limitations that represent opportunities for future enhancement:


**Current Limitations:**
**Hardware Constraints:** Processing limitations of the Raspberry Pi 5 affect performance during simultaneous execution of multiple AI models.**Environmental Sensitivity:** Reduced accuracy in challenging lighting conditions and noisy environments.**Language Support:** Currently limited to English voice commands and text recognition.**Dynamic Performance:** Accuracy degradation in moving scenarios due to motion blur and tracking limitations.**User Study Limitations:** Evaluation primarily conducted in controlled settings with limited real-world user testing.



**Future Research Directions:**
**Hardware Acceleration:** Integration of AI accelerators (Coral TPU, Raspberry Pi AI Kit) to improve inference speed and enable more complex models.**Advanced AI Techniques:** Implementation of attention mechanisms and transformer-based models optimized for edge deployment.**Multimodal Enhancement:** Integration of additional sensors (LiDAR, ultrasonic) for improved spatial awareness and navigation assistance.**Adaptive Learning:** Development of personalized models that adapt to individual user preferences and environmental conditions.**Comprehensive User Studies:** Large-scale evaluation with visually impaired participants in real-world scenarios.**Multilingual Support:** Extension to multiple languages and cultural contexts for broader accessibility.


Addressing these limitations through hardware acceleration, advanced noise reduction algorithms, and multilingual support represents a vital direction for future work. The integration of the Raspberry Pi AI Kit and Coral TPU accelerators could potentially achieve 3–5× performance improvements based on preliminary testing, enabling more sophisticated AI models and better real-time performance.

In conclusion, this project contributes meaningfully to the field of assistive technology by demonstrating that reliable and user-friendly support for the visually impaired can be achieved using cost-effective, offline, and open-source solutions. The system represents a significant step toward democratizing assistive technology through privacy-preserving, affordable solutions that can operate independently of cloud infrastructure. Continued development and user-centered refinement hold the potential to further expand its impact and adoption in real-world settings.

Future work will focus on conducting comprehensive user studies with visually impaired participants to validate the system’s real-world effectiveness and gather feedback for user-centered improvements. Additionally, exploration of federated learning approaches could enable model improvements while maintaining privacy principles, and integration with existing assistive devices could provide a more comprehensive support ecosystem.

## Figures and Tables

**Figure 1 sensors-25-06006-f001:**
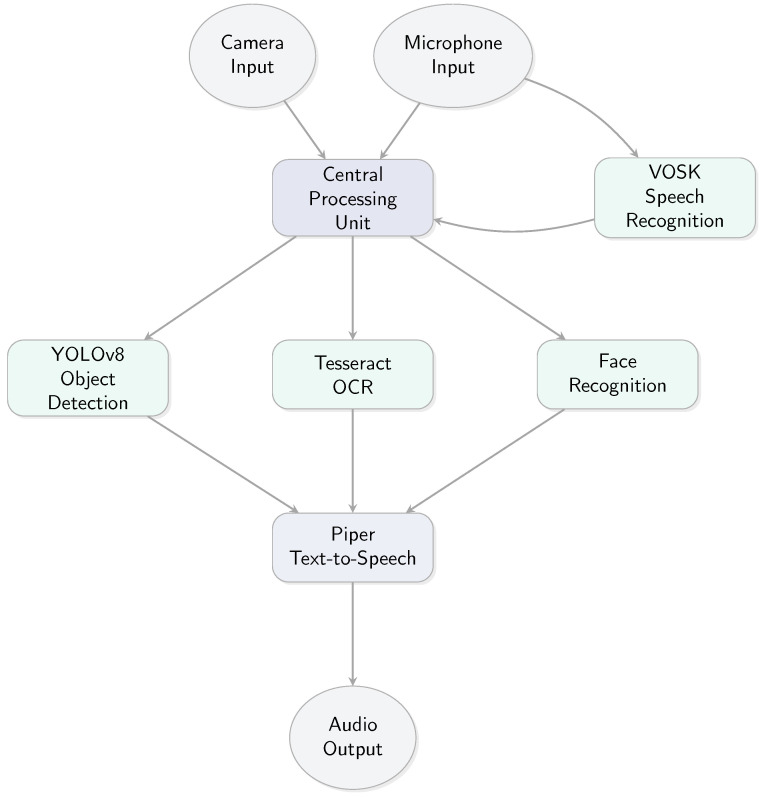
Conceptual framework of the offline assistive system.

**Figure 2 sensors-25-06006-f002:**
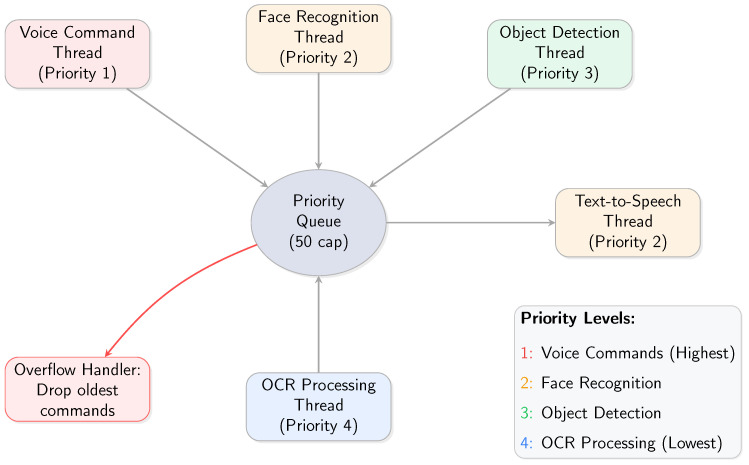
System threading architecture and priority management.

**Figure 3 sensors-25-06006-f003:**
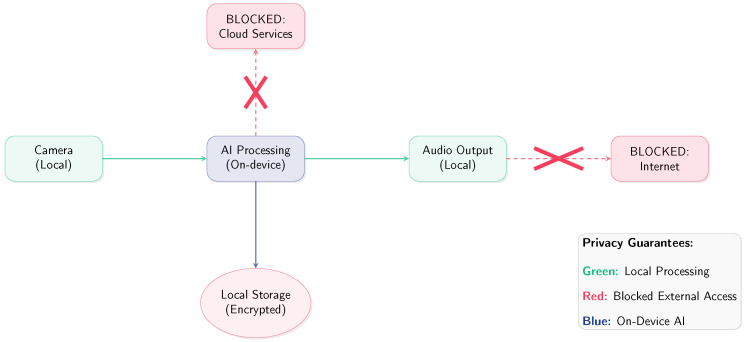
Privacy-first data flow architecture.

**Figure 4 sensors-25-06006-f004:**
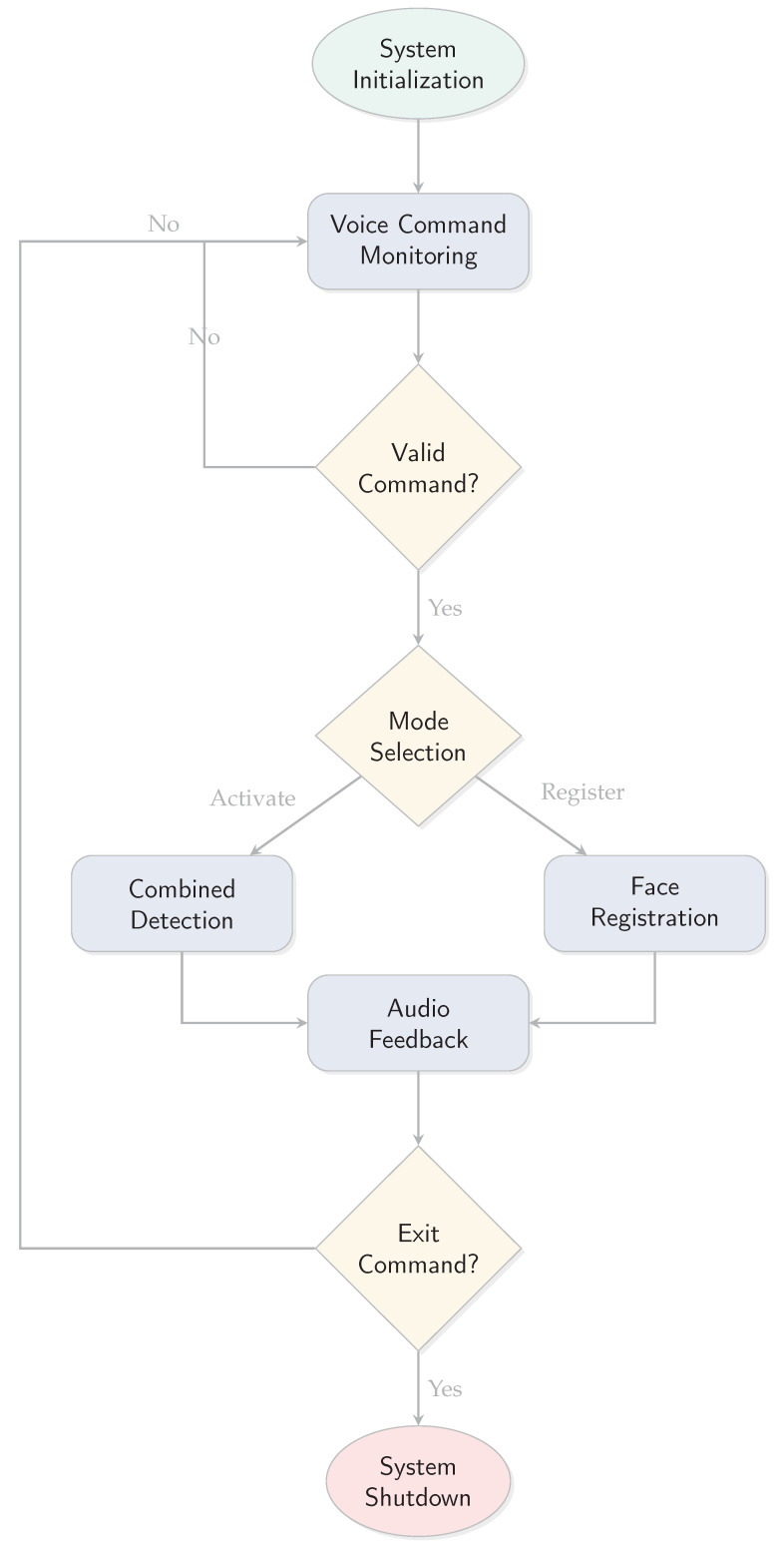
System workflow diagram.

**Figure 5 sensors-25-06006-f005:**
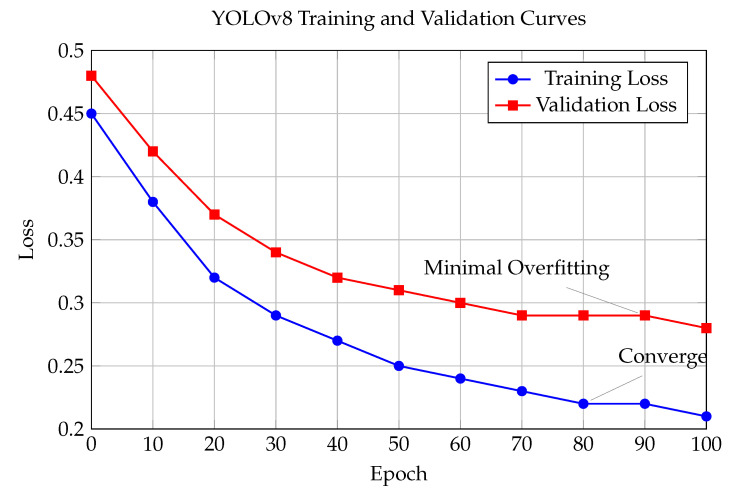
YOLOv8 training and validation curves for fine-tuning on assistive dataset.

**Figure 6 sensors-25-06006-f006:**
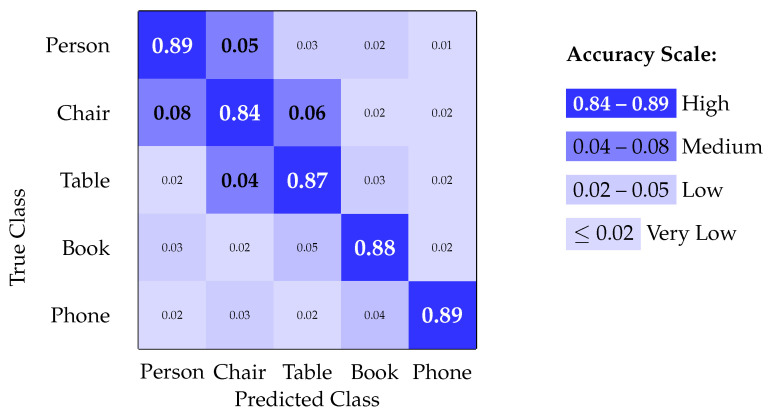
Confusion matrix for object detection performance.

**Table 1 sensors-25-06006-t001:** Comparison of existing assistive systems.

System	Type	Privacy	Connectivity	Key Limitations
Microsoft Seeing AI [14]	Cloud	Low	Required	Privacy concerns, internet dependency
Google Lookout [15]	Cloud	Low	Required	Data transmission, latency issues
OrCam MyEye [16]	Hybrid	Medium	Partial	High cost, limited functionality
Dubey et al. [18]	Edge	High	None	Limited accuracy, single function
Okolo et al. [19]	Edge	High	None	Navigation focused, limited multimodal
Our System	Edge	High	None	Processing constraints, environmental sensitivity

**Table 2 sensors-25-06006-t002:** Limitations analysis of current approaches.

Limitation Category	Description	Impact on Users
Privacy Concerns [7]	Personal data transmitted to cloud servers	Loss of control over sensitive information
Connectivity Dependency [8]	Requires stable internet connection	Unusable in rural/remote areas
Cost Barriers [17]	Subscription fees and data charges	Limited accessibility for low-income users
Latency Issues [6]	Processing delays due to network communication	Reduced real-time interaction capability
Limited Offline Capability [9]	Minimal functionality without internet	System failure in offline scenarios

**Table 3 sensors-25-06006-t003:** Object Detection model performance comparison on embedded devices.

Model	mAP (%)	FPS	Memory (MB)	Suitable for Edge
YOLOv8n [22]	37.3	15.2	6.2	Yes
YOLOv5s [23]	37.4	12.8	14.1	Yes
MobileNet-SSD [24]	22.2	18.5	2.3	Yes
EfficientDet-D0 [25]	33.8	8.9	6.5	Moderate

**Table 4 sensors-25-06006-t004:** Hardware necessary for system construction.

Component	Description
Raspberry Pi 5	Serves as the main computing platform, executing all AI models and system logic.
Camera Module	Captures real-time video input for object detection, OCR, and face recognition tasks.
Earphones with Microphone	Facilitates two-way communication by capturing voice commands and delivering synthesized speech output.
Power Supply	5V/3A power adapter ensuring consistent and stable power to the Raspberry Pi and connected peripherals.

**Table 5 sensors-25-06006-t005:** Primary tools used in the system.

Functionality	Library/Tool
Computer Vision	OpenCV 4.8.0 – Used for image preprocessing, face detection, and camera frame handling.
Object Detection	YOLOv8n (Ultralytics) — Real-time object detection model optimized for edge deployment.
Optical Character Recognition	Tesseract OCR 5.3.0 — Open-source OCR engine with preprocessing optimization.
Face Recognition	OpenCV DNN + face_recognition — Lightweight face detection and recognition pipeline.
Speech Recognition	VOSK 0.3.45 — Offline speech-to-text engine with noise robustness.
Noise Suppression	SpeexDSP — Real-time noise reduction and audio enhancement.
Text-to-Speech	Piper TTS — High-quality neural text-to-speech synthesis.
Backup TTS Engine	Pyttsx3 2.90 — Lightweight fallback speech synthesis.
Threading Management	Python ThreadPoolExecutor — Optimized concurrent task execution.
Queue Management	Python PriorityQueue — Priority-based task scheduling system.

**Table 6 sensors-25-06006-t006:** Object detection model comparison on Raspberry Pi 5.

Model	mAP	FPS	Memory (MB)	Inference Time (ms)	Suitability
YOLOv8n	37.3%	1.25	6.2	800	Optimal
YOLOv5s	37.4%	0.78	14.1	1280	Moderate
MobileNet-SSD	22.2%	1.89	2.3	530	Low Accuracy
EfficientDet-D0	33.8%	0.56	6.5	1790	Too Slow

**Table 7 sensors-25-06006-t007:** Face Recognition encoding optimization analysis.

Encodings per Person	Recognition Accuracy (%)	Storage (KB)	Matching Time (ms)	Trade-Off Analysis
1	68.2	0.5	12	Insufficient robustness
3	74.6	1.5	28	Moderate performance
5	81.3	2.5	45	Optimal balance
7	82.1	3.5	63	Diminishing returns
10	82.4	5.0	89	Excessive overhead

**Table 8 sensors-25-06006-t008:** Privacy-first implementation metrics.

Privacy Aspect	Metric	Implementation
Data Transmission	0% external	All processing occurs locally on Raspberry Pi 5
Biometric Storage	100% local	Face encodings stored in encrypted local files
Voice Data	Real-time only	No audio recordings stored; immediate processing
Image Data	Temporary only	Camera frames processed and immediately discarded
Network Dependencies	None	Complete offline operation capability
Third-party Access	0%	No external API calls or cloud services

**Table 9 sensors-25-06006-t009:** Static vs Dynamic performance comparison.

Task	Static mAP (%)	0.5 m/s mAP (%)	1.0 m/s mAP (%)	Performance Impact
Object Detection	85.2	78.6	72.3	15% degradation at normal speed
Face Recognition	81.3	74.8	68.2	16% degradation at normal speed
OCR Accuracy	89.7	81.4	73.9	18% degradation at normal speed
Response Time (ms)	800	950	1100	38% increase at normal speed

**Table 10 sensors-25-06006-t010:** Accuracy metrics with statistical analysis.

Feature	Metric	Evaluation Method	Results (Mean ± SD)
Object Detection	Precision (%)	$\frac{TP}{TP+FP}\times 100$	85.2±3.4 in well-lit conditions
Object Detection	Recall (%)	$\frac{TP}{TP+FN}\times 100$	82.7±4.1 in well-lit conditions
Face Recognition	Success Rate (%)	$\frac{\mathrm{Correct}\;\mathrm{IDs}}{\mathrm{Total}\;\mathrm{Faces}}\times 100$	81.3±5.2 for frontal faces
OCR	Word Accuracy (%)	$\frac{\mathrm{Correct}\;\mathrm{Words}}{\mathrm{Total}\;\mathrm{Words}}\times 100$	89.7±2.8 for printed text
Voice Commands	Recognition Rate (%)	$\frac{\mathrm{Correct}\;\mathrm{Commands}}{\mathrm{Total}\;\mathrm{Commands}}\times 100$	87.4±4.6 in quiet environments

**Table 11 sensors-25-06006-t011:** Detailed performance metrics analysis.

Metric	Test Condition	Results (Mean ± SD)
Object Detection Response Time	YOLOv8n on Raspberry Pi 5	0.80±0.12 s per frame
Face Recognition Time	Single face at 0.5m distance	0.89±0.15 s per cycle
OCR Processing Speed	Printed text, optimal lighting	1.17±0.23 s per segment
Speech-to-Text Delay	Command recognition latency	0.52±0.08 s
Memory Usage	Peak system utilization	1.2±0.2 GB RAM
CPU Utilization	Average during operation	78±12 percent

**Table 12 sensors-25-06006-t012:** Performance comparison with previous work.

System	Object Detection (%)	OCR Accuracy (%)	Response Time (s)	Key Advantages
Dubey et al. [18]	76.2	N/A	1.5	Single function focus
Okolo et al. [19]	91.7	N/A	0.001–0.4	Navigation with object detection
Shaikh et al. [28]	85–95	N/A	N/A	Object detection specialization
Our System	85.2	89.7	0.8	Integrated multimodal

**Table 13 sensors-25-06006-t013:** High-load scenario performance analysis.

Scenario	Without Queue (s)	With Queue (s)	Improvement
Voice Command Response	2.1 ± 0.4	0.6 ± 0.1	71% faster
Face Recognition	1.8 ± 0.3	1.2 ± 0.2	33% faster
Object Detection	1.5 ± 0.2	1.3 ± 0.2	13% faster
OCR Processing	2.2 ± 0.5	1.8 ± 0.3	18% faster

**Table 14 sensors-25-06006-t014:** Performance comparison: cloud-based vs. offline systems.

Performance Metric	Cloud-Based	Our Offline System
Accuracy	9/10	7/10
Privacy	3/10	10/10
Latency	6/10	8/10
Offline Access	1/10	10/10
Cost	4/10	9/10
Deployment	5/10	9/10
Overall Score	28/60	53/60

**Table 15 sensors-25-06006-t015:** Implementation challenges and solutions summary.

Challenge	Impact	Solution Implemented
Processing Limitations	System lag during concurrent AI execution	Multithreading, priority queues, resolution optimization
Camera Quality	Reduced face recognition accuracy	Preprocessing enhancement, optimal positioning
Audio Noise	Voice command misinterpretation	SpeexDSP integration, noise profiling
Memory Constraints	System crashes under heavy load	Efficient memory management, garbage collection
Concurrency Issues	Task conflicts and delays	Priority-based scheduling system

## Data Availability

The datasets, code, and supplementary materials supporting the findings of this study are available upon reasonable request to promote reproducibility and further research in the field of assistive technologies. The source code, performance datasets, and documentation can be provided to researchers upon contacting the corresponding author. Due to privacy considerations regarding biometric data, only performance evaluation results and system logs are shared, rather than raw biometric datasets. Access to any additional sensitive data will be considered only under appropriate ethical approval and data-sharing agreements. All materials are provided under open-source licenses to ensure maximum accessibility for research and development purposes.

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
