# Peer review of "Development of a Fully Autonomous Offline Assistive System for Visually Impaired Individuals: A Privacy-First Approach"

_sensors, 2025, doi:10.3390/s25196006_

Round 1

Reviewer 1 Report

Comments and Suggestions for Authors

Good to see the research topic. However, I have the following concern related to your paper:

  1. In introduction chapter, I can not see any relevant reference.
  2. Literature section is still too poor. To understand the research gaps, you should add a summary tables highlighting limitations of existing papers/models
  3. In methodology section, it is wonder why did you select Yolov8? I am very much interested to see it, people usually select light weight architectures or high accuracy models, I did not get the reason of using YoloV8.
  4. You should compare YoloV8 model with other segmentation based methods
  5. There are so many chapters(8), you should merge additional sections under sub-sections like 5
  6. You should report training curve, loss curve, confusion metrics and other performance metrics
  7. Limitations and future scopes you should merge with conclusion chapter

Author Response

Dear Reviewer 1,

Thank you for your thorough evaluation and constructive feedback on our manuscript. We have carefully addressed each of your comments and made substantial revisions to improve the manuscript quality, structure, and technical depth.

Please see the attachment for our detailed point-by-point response to your comments, including the specific changes made in the revised manuscript.

We sincerely appreciate your time and expertise in reviewing our work.

Best regards,
Mohammad Fayez Al Bataineh (Corresponding Author)

Reviewer 2 Report

Comments and Suggestions for Authors

This paper focuses on the environmental interaction needs of visually impaired people in the scenario of "no network dependence and privacy protection". Based on Raspberry Pi 5, a multimodal assistance system integrating real-time object detection, optical character recognition, face recognition, and offline voice control is constructed. Through open source tool integration and lightweight optimization, fully offline operation of hands free interaction is achieved. The research topic is closely related to the actual pain points of the visually impaired population, and the technical route combines theoretical innovation and engineering practicality. It has verified a detection/recognition accuracy of 75% -90% and sub second response time in a controlled indoor environment, providing valuable solutions for visually impaired assistive technology in low resource scenarios.

I suggest publishing this paper after minor revisions, with the following revisions:

(1) The system uses a multi-threaded joint FIFO queue to manage concurrent tasks, but does not specify the core scheduling rules: thread priority allocation (such as whether voice command processing takes priority over object detection), queue capacity limit, and overflow handling mechanism (such as whether to discard low priority tasks when commands pile up). Suggest the author to supplement the thread architecture diagram and explain the basis for priority setting (such as setting voice commands as the highest priority based on the real-time interaction needs of visually impaired users); Simulate high load scenarios (such as simultaneous triggering of object detection, OCR, and facial recognition) and compare the system response delay with and without queue scheduling.

(2) The experiment is mainly based on static scenes, and the adaptability to dynamic scenes has not been verified, such as tracking performance when the target is moving and text dynamic capture. Suggest the author to construct a dynamic test set (such as recording real-time videos of visually impaired users walking and annotating moving targets), compare the mAP differences in object detection in static/dynamic scenes, test the system response delay at different moving speeds (such as 0.5m/s/1m/s), and analyze the impact of dynamic blur on OCR/object detection.

(3) To make the introduction more comprehensive, it is recommended that the author refer to the following two papers on algorithms. 1)Towards visual interaction: hand segmentation by combining 3d graph deep learning and laser point cloud for intelligent rehabilitation. 2)Intelligent rehabilitation in an aging population: empowering human-machine interaction for hand function rehabilitation through 3d deep learning and point cloud.

(4) The paper chooses YOLOv8 as the core model for object detection because it balances real-time performance and accuracy. However, from the perspective of embedded vision theory, YOLOv8's cross window attention mechanism is theoretically more efficient than MobileNet's deep separable convolution on low computing hardware such as Raspberry Pi 5. Why was a lighter MobileNet series model not chosen, and how should the differences in theoretical adaptability be demonstrated.

(5) Facial recognition improves recognition accuracy from different angles through multi pose encoding storage. From the perspective of biometric recognition theory, storing multiple sets of encoding can improve robustness, but it will increase local storage overhead and matching time. In theory, how to determine the threshold for encoding quantity to balance recognition accuracy, storage cost, and matching efficiency.

Author Response

Dear Reviewer 2,

Thank you for your thorough evaluation and constructive feedback on our manuscript. We have carefully addressed each of your comments and made substantial revisions to improve the manuscript quality, structure, and technical depth.

Please see the attachment for our detailed point-by-point response to your comments, including the specific changes made in the revised manuscript.

We sincerely appreciate your time and expertise in reviewing our work.

Best regards,
Mohammad Fayez Al Bataineh (Corresponding Author)

Reviewer 3 Report

Comments and Suggestions for Authors

This paper explores the integration and potential of open-source AI models in developing a fully offline assistive system that can be locally set up and operated to support visually impaired individuals.

However, there are observations the authors have to put into Conferderation to enhance the quality of the manuscript. 

Title: "Development of a Fully Autonomous Offline Assistive System for Visually Impaired Individuals: A Privacy-First Approach". The author did not demonstrate the "Privacy-First Approach" in the manuscript. Authors are advice to create a subsection and demonstrate: What, how and results of Privacy-First Approach with standard metrics and results. Or else consider removing it from the title. 

Abstract: Although is good but can be improved by adding the introduction and problem statement at the begin. Both components are currently missing in the abstract. 

  1. Introduction:The introduction should be beef up with more citations. This enables your work to have its problem statement established in literature. 
  2. Related WorkThe citation should be sequential in the body text and must be referenced.  e.g. [1], [2], [3] and so on. There should be a table summary or comparing others and your work using the strong of your work as comparative advantage. More related works have to being reviewed to establish the rationale behind this study and please not recycling same material. 
  3. System Design and Methodology The conceptual framework of this study should come up early enough in sub-section 3.1 to enhance reader understanding. e.g. Figure 1. More diagrams such Sequence, Activity, and Deployment etc. in different aspect of the methodology and thereafter explained currently been done by the authors. All bullet points should be changed either (i), (ii), etc. or (a), (b) etc. Algorithms use should be correctly labelled and refers to such. Section 4 Data processing and optimization should be a subsection of Section 3
  4. Testing and Evalution Table 3: Accuracy metrics; all stated metrics and their related equations should state and correctly cited and labelled. There should be a table to compare your results with previous related work done to validate your work using the standard metrics and justification for your results. Figure is not clear and should be clearly explained. 
  5. Acknowledgement should be moved to after Conclusion section.

Author Response

Dear Reviewer 3,

Thank you for your thorough evaluation and constructive feedback on our manuscript. We have carefully addressed each of your comments and made substantial revisions to improve the manuscript quality, structure, and technical depth.

Please see the attachment for our detailed point-by-point response to your comments, including the specific changes made in the revised manuscript.

We sincerely appreciate your time and expertise in reviewing our work.

Best regards,
Mohammad Fayez Al Bataineh (Corresponding Author)

Round 2

Reviewer 1 Report

Comments and Suggestions for Authors

Thank you for addressing the comments. I do not have any further comments. I am expecting you to add a data availability statement along with a GitHub link with sample codes 

Reviewer 2 Report

Comments and Suggestions for Authors

The author has provided excellent answers to the questions raised by the reviewer, and this paper is acceptable.

Reviewer 3 Report

Comments and Suggestions for Authors

The authors have made most of the corrections and the manuscript has greatly improved. However, there are still some observations that were done either completely or very well. 

 Line 70-71:  "the first comprehensive offline multimodal assistive system integrating object detection, OCR, face recognition, and voice control on a single  edge device." This statement is too authoritative. Especially if the authors have not gone through all database both online and offline, published and unpublished, also all through languages to ascertain that such has not be done. Rather the authors should use a more research friendly words like a "novel, etc."

 In "Table 1. Comparison of Existing Assistive Systems [18]" all items should be cited individually using their primary source like items 1-4 on the table. Remove the secondary source [18] on the title. Also, all item on "Table 2. Limitations Analysis of Current Approaches" should be cited and referenced. 

All figures and table must be cited in the body text before presentations. 

Authors should ensure all citations on the body of the text must be referenced. 

Reference "[1] World Health Organization. World Report on Vision, 2019." should include source URL and date accessed

References [4] Wang, X.; et al. and [9] Kong, L.; et al.  should have their authors written all like other references. 

All references should have their DOI, while the once from website should have their URLs and date accessed. 
